# To Fiber or Not to Fiber: The Swinging Pendulum of Fiber Supplementation in Patients with Inflammatory Bowel Disease

**DOI:** 10.3390/nu15051080

**Published:** 2023-02-21

**Authors:** Natasha Haskey, Stephanie L. Gold, Jeremiah J. Faith, Maitreyi Raman

**Affiliations:** 1Department of Biology, The Irving K. Barber Faculty of Science, University of British Columbia—Okanagan, 3187 University Way, Kelowna, BC V1V 1V7, Canada; 2Division of Gastroenterology, Cumming School of Medicine, University of Calgary, 6D33 TRW Building, 3280 Hospital Drive NW, Calgary, AB T2N 4N1, Canada; 3Division of Gastroenterology, Icahn School of Medicine at Mount Sinai, 1 Gustave L. Levy Place, New York, NY 10029, USA; 4Precision Immunology Institute and Department of Genetics and Genomic Sciences, Icahn School of Medicine at Mount Sinai, 1 Gustave L. Levy Place, New York, NY 10029, USA

**Keywords:** inflammatory bowel disease, dietary fiber, prebiotic, novel fiber sources

## Abstract

Evidence-based dietary guidance around dietary fiber in inflammatory bowel disease (IBD) has been limited owing to insufficient reproducibility in intervention trials. However, the pendulum has swung because of our increased understanding of the importance of fibers in maintaining a health-associated microbiome. Preliminary evidence suggests that dietary fiber can alter the gut microbiome, improve IBD symptoms, balance inflammation, and enhance health-related quality of life. Therefore, it is now more vital than ever to examine how fiber could be used as a therapeutic strategy to manage and prevent disease relapse. At present, there is limited knowledge about which fibers are optimal and in what form and quantity they should be consumed to benefit patients with IBD. Additionally, individual microbiomes play a strong role in determining the outcomes and necessitate a more personalized nutritional approach to implementing dietary changes, as dietary fiber may not be as benign as once thought in a dysbiotic microbiome. This review describes dietary fibers and their mechanism of action within the microbiome, details novel fiber sources, including resistant starches and polyphenols, and concludes with potential future directions in fiber research, including the move toward precision nutrition.

## 1. Introduction

Inflammatory bowel disease (IBD) affects over six million people worldwide, with rates in North America reaching approximately three million [1]. The pathogenesis of Crohn’s disease (CD) and ulcerative colitis (UC) remains unclear; however, evidence suggests that it results from a complex interaction between genetic risk factors, an aberrant host immune response, alterations in the gut microbiome, and environmental factors, including diet [2]. The adoption of Westernized eating habits has led to a significant reduction in fiber consumption and is linked to an increased prevalence of digestive diseases such as IBD, partially through alterations in microbial composition and changes in the epithelial barrier [3]. In particular, alterations in the abundance of bacterial species that metabolize non-digestible dietary fiber (e.g., *Faecalibacterium prausnitzii, Roseburia* spp., and *Eubacterium rectale*) are associated with IBD [3]. Fiber-degrading microbes produce important metabolites, such as short-chain fatty acids (SCFAs), that regulate metabolic and immune homeostasis and gut barrier integrity [4]. Due to the interest in the emerging role of the gut microbiome in health and the potential role dietary fiber plays in altering the gut microbiome, there is a renewed interest in manipulating dietary fiber as a therapeutic target to manage IBD [3].

A growing body of evidence supports the benefit of dietary fibers in maintaining a health-associated microbiome in the absence of IBD [5,6,7], with much of this knowledge extrapolated to the IBD patient population. Very little is known about which fibers are optimal and in what form/subtypes and quantity they should be consumed to benefit patients with IBD [8]. Additionally, as fiber degradation depends on the availability of host microbes, more research is needed to determine how an altered gut microbiome and disease state (active disease or remission) may impact fermentation patterns in IBD [8]. Therefore, this review describes dietary fibers and their mechanism of action within the microbiome, details novel fiber sources, including resistant starches and polyphenols, and concludes with potential future directions in fiber research, including the move toward precision nutrition.

## 2. Key Concepts in Fiber and IBD

### 2.1. What Is Fiber?

While many studies lump various types of “fibers” together, it is essential to recognize that dietary fibers are heterogeneous substances with each fiber having varied biological effects [9]. The definition of fiber has been debated for many years; however, in 2009, a formal definition of fiber was published by the Codex Alimentarius Commission, which described fiber as “carbohydrate polymers with ten or more monomer units that are not hydrolyzed by endogenous enzymes in the human small intestine” [10]. The definition of fiber varies among countries, as the decision to include carbohydrate monomeric unit counts of 3–9 is left to the decision of national authorities [10]. In essence, dietary fibers are not degraded by host enzymes; therefore, escape digestion in the small bowel, and pass into the large bowel intact, where they undergo partial or complete anaerobic fermentation by the microbiota [11].

Dietary fibers are commonly divided by subtype based on solubility, viscosity, and fermentation properties, with health benefits highly correlated with these physical attributes (Figure 1) [11,12]. Broadly speaking, depending on the solubility of the fiber in water, it is classified as insoluble or soluble. Soluble fibers have a water-holding capacity with high viscosity/gel-forming properties and are readily fermented by the microbiota, including pectins, arabinoxylans, beta-glucans, and water-soluble gums [11]. Dietary sources of these fibers include whole grains (e.g., oats, barley), legumes, the flesh of fruit and vegetables, and seeds (e.g., flax seeds or chia seeds [11]. Psyllium is a soluble fiber with high viscosity/gel-forming properties; however, it is poorly fermented by the microbiota [13]. In contrast, insoluble fibers, that lack water-holding capacities, such as cellulose, hemicellulose, and lignan, are less fermentable by the bacteria [11]. These fibers are typically found in whole wheat bread, pasta, fruit and vegetable skins, nuts, and seeds [11].

Prebiotics are defined as a substrate that is selectively utilized by host microorganisms conferring a health benefit [14]. Unlike dietary fibers such as pectins, cellulose, and xylans, which are metabolized by a wide variety of gut microorganisms, prebiotics are metabolized by specific health-associated microorganisms (e.g., *Lactobacillus* spp. and *Bifidobacterium* spp.) [14]. The most well-researched prebiotics include inulin, FOS, and GOS, whereas other fermentable carbohydrates (e.g., lactulose, resistant starch) are “candidate prebiotics” [14].

The current definition of dietary fiber is problematic, considering that fiber is a substrate for metabolism by gut microbes. Although not all dietary fibers are prebiotics, all prebiotics are a form of dietary fiber. While some soluble fiber sources (i.e., psyllium) are poorly fermented by microbes, inulin, fructooligosaccharides (FOS), galactooligosaccharides (GOS), and wheat dextrin are readily fermented by the microbiota [14]. However, because of their lack of water-holding capacity, they are categorized as insoluble fibers. To rectify this discrepancy, the term “microbiota-accessible carbohydrates”(MACs) has been proposed; these are defined as any carbohydrate or food product resistant to stomach degradation, not absorbed by the small bowel, and can be fermented or metabolized by the host microbes [15]. MACs can be found in plant products, and the number of MACs present in a food item will vary from person to person, as the microbial breakdown of these foods depends on the microbial profile of each individual [15]. Both clinical and murine models have demonstrated that without the microbiota to metabolize a specific food item, the host will not get the same benefits from the food source compared to others whose microbiomes can metabolize the dietary MACs [15]. Therefore, MACs should not be viewed as a static characteristic of specific dietary components and instead represent the potential metabolic activity associated with carbohydrates in a particular microbiome [15]. 

Finally, while most dietary fiber intervention studies have focused on isolated, single fiber or fiber extracts, this is not how we generally consume dietary fiber [16,17]. Plant-based foods such as vegetables, fruit, nuts, seeds, legumes, and grains are whole foods that are not just one single source or extract of fiber but contain a complex three-dimensional plant cell matrix (i.e., plant cell walls), termed “intrinsic fibers” [18]. Within the plant cell walls, various types of fiber are stored in vacuoles (e.g., starch, fructans, sugars, phytochemicals) which differ according to the plant source. As a result, the three-dimensional plant cell matrix has important consequences for the microbiota to access the individual fibers which not only influence digestion but also fermentation patterns [19]. There is a paucity of human clinical trials that have examined the impact of intrinsic fibers and the gut microbiome in both healthy individuals and those living with IBD (reviewed by Pulhmann et al., 2022) [19].

### 2.2. Fiber Guidelines Are Changing

Over the years, evidence-based dietary guidance around fiber and IBD has been limited owing to insufficient reproducibility in intervention trials [20,21]. However, the pendulum has recently swung because of our increased understanding of the importance of fiber in maintaining a health-associated microbiome in healthy individuals [5,6,7]. Meta-analyses of observational studies demonstrate a significant inverse association between higher fiber intake (>22 g/day) and risk of developing CD [HR 0.59, 95% CI 0.39–0.90] but not UC [3,21].

Existing intervention trials have shown that dietary fibers can improve IBD symptoms [20], increase short-chain fatty acid production [4], alter the microbiota [22], enhance health-related quality of life [23] and balance inflammation [12]. A recent meta-analysis reported a significant inverse association between higher vegetable consumption in UC (OR = 0.71) and higher fruit consumption (both UC and CD), and the risk of new IBD onset [3]. 

For decades, the most prominent dietary recommendation for patients with IBD has been to follow a low-fiber or low-residue diet, especially when the disease is active, to minimize symptoms [24]. In this context, fiber, as a general term, was traditionally thought of as mainly a bulking agent that increased stool frequency and volume, so if one was experiencing symptoms such as diarrhea, removing fiber from the diet was thought to alleviate the symptoms. Although “low fiber” and “low residue” are often used interchangeably, these terms are distinct. A low-fiber diet restricts foods high in all fibers, whereas a low-residue diet limits fiber to 10–15 g/day and other foods that could increase malabsorption (e.g., lactose in dairy products) [24]. Ironically, there is limited research to support the use of either of these diets in IBD patients. A prospective dietary study published in 1985, compared a low residue diet (no fruits or vegetables other than a banana and peeled potatoes, no dairy, no whole grains or legumes) to an unrestricted “normal Italian diet” in patients with non-structuring CD [25]. Ironically, the study found no difference in clinical outcomes (symptoms, hospitalization rate, need for surgery, complications, or postoperative recurrence) after following the diet for 29 months [25]. The guidelines are based on anecdotal evidence, with patients reporting an improvement in symptoms when following a diet reduced in fiber that is less than 10 g/day [24].

The “swing in the pendulum” commenced in 2022 when the British Dietetic Association consensus guidelines proposed that fiber restriction was not required in those with stable or quiescent IBD [26]. Moreover, in this recent guideline, the recommendations about fiber intake in CD patients with structuring phenotype have become somewhat more nuanced; patients are still recommended to avoid fibrous foods such as tough outer skins of fruits and vegetables, tough meats, etc.; however, they are encouraged to include foods rich in soluble fibers when consumed with adequate hydration [26]. Similarly, the European Society for Clinical Nutrition and Metabolism (ESPEN) 2023 practical guidelines on Clinical Nutrition in IBD; instead of recommending a “low fiber” or “low residue” diet for patients with small bowel strictures, they recommended a diet with “adapted textures” (e.g., soft, cooked, and peeled vegetables, and soft, or peeled fruit pureed in a smoothie) [27]. While avoiding insoluble fibers in those with strictures is a “logical approach,” no robust data supports this practice. Despite a lack of clear consensus on the optimal amount, type, and even preparation of fiber in the diet for patients with IBD, the gastrointestinal community is slowly moving away from the generalized recommendation to avoid fiber. 

### 2.3. Mechanisms of Action of Dietary Fiber

The gut microbiome plays a fundamental role in human health and diseases that exist in a tripartite relationship between the microbiota, epithelial barrier, and immune system [28] (Figure 2). This complex ecosystem provides essential life functions, including an intermediary role in synthesizing B vitamins and vitamin K, maintaining immune homeostasis, and producing key metabolites [4,28,29,30]. In IBD, this complex ecosystem is altered. As such, the working hypotheses are that the disease is driven by microbial dysbiosis, impaired epithelial barrier function, and defects in the protective mucus layer, which drive a pro-inflammatory state [31].

Several studies have identified variations in the microbial taxa associated with IBD [31,32,33]; however, in comparison to healthy controls, health-associated microbes such as *Clostridium* groups IV and XIVa spp., *Bacteroides* species (spp.), *Roseburia* spp., *Bifidobacterium* spp. and *F. prausnitzii*, known as key producers of the SCFA butyrate, are frequently reduced in active IBD [31,32], along with an increased abundance of pathobionts, including select species of *Fusobacteriaceae* [33], as well as *Escherichia*, and *Desulfovibrio*, from the phylum Proteobacteria [31,32]. Prebiotic consumption (e.g., inulin/FOS combination) in mild-moderately active and quiescent CD can modulate the composition of the microbiota, mainly through enrichment of the *Bifidobacterium* spp. and *Lactobacillus* spp., increasing the production of SCFA and through pathogen exclusion [34]. Dietary patterns might be more important than the consumption of individual types of fiber in supporting specific taxa, as increased adherence to a Mediterranean diet (containing high levels and diverse forms of fiber) has been associated with increased abundances of *F. prausnitzii* and *Roseburia* spp. [35]. 

The intestinal epithelium is a physical barrier separating the lumen from the innermost layer of the gut. The mucus layer provides a barrier between the microbes and hosts immune cells and is situated on top of intestinal epithelial cells. The preservation of the mucus layer, which separates the gut microbiota from the intestinal epithelium, has consistently been identified as a critical component in preserving a healthy intestinal barrier [36,37]. The mucus layers consist of highly glycosylated proteins (mucins), with mucin 2 being one example secreted by goblet cells. Mucin degradation and turnover are essential to protect the mucosal barrier, as they maintain a balance of equilibrium between the biosynthesis of mucosal membranes and the secretion and breakdown of the mucus [38]. Barrier defects have been observed in colonic biopsies of patients with UC, particularly alterations in mucin activity, expression, synthesis, and structure [38]. A low-fiber diet promotes the expansion of mucus-layer-degrading bacterium altering epithelial barrier function, increasing its permeability, and resulting in the translocation of pathogens [39]. Adding specific dietary fiber combinations can restore the mucus layer demonstrating one potential mechanism for the benefit of fiber in IBD [39]. *Akkermansia muciniphila*, often found in lower abundance in UC, is the most well-studied mucus-layer-degrading bacterium [32]. Although its function also involves the degradation of the mucus layer, it has the ability to convert the mucin in the host to beneficial products, such as SCFAs [32]. Other mucin-degrading bacterial strains include *Clostridium* spp., *Ruminococcus* spp., *Bacteroides* spp., *Prevotella* spp., and *Bifidobacterium* spp. [40]. 

In a health-associated microbiome, dietary fiber is proposed to be a regulator of gut homeostasis and can impact the immune system directly and indirectly through SCFAs [4,28]. Immune, epithelial, and adipocyte cells contain G-protein coupled receptors (GPCRs) that can bind SCFAs and induce changes in cytokine levels and various signaling pathways producing pro and anti-inflammatory effects [41]. Butyrate, the primary fuel source for colonocytes, also modulates gut barrier function through tight junction protein assembly, goblet cell activation, and function, as well as epithelial cell growth [41]. These functions are regulated transcriptional co-factors that regulate gene expression (e.g., histone deacetylases (HDACs)) [4,28]. NF-κB is a transcription factor that plays a key role in the regulation of the inflammatory response through upregulating pro-inflammatory genes. SCFAs, such as butyrate can modulate epithelial cell, macrophage, and dendritic cell cytokine and chemokine secretion, by inhibiting NF-κB activity and increasing the transcription of the peroxisome proliferator-activated receptor gamma (PPARγ) [42]. Finally, SCFAs can directly affect and influence immune homeostasis by activating pattern recognition receptors (PRRs) such as C-type lectin receptors or Toll-like receptors (mainly TLR-2 and TLR-4) found on epithelial cells and cells of the innate immune system (reviewed elsewhere by Cai et al. [28].

### 2.4. Lessons from Mouse Models

Controlled genetics and environmental conditions of mouse models provide the opportunity to discover mechanisms and principles that can be applied to human studies. In gnotobiotic mouse models with defined bacterial consortia, dietary perturbations have a rapid (24 h) and consistent influence on the relative abundance of gut microbiome strains [43,44]. A specific diet drives the microbiota to a given state of relative abundance for each strain that is consistent, repeatable, and independent of the order in which diets are provided. A rodent study using mice with dextran sulfate sodium (DSS)-induced colitis identified dietary fiber as the most beneficial dietary perturbation in preventing colitis, with a large variation in benefit across fiber types, suggesting that specific dietary fiber manipulations could have therapeutic potential in IBD [45]. Animal models with monotonously defined diets have the potential to identify mechanisms that can be tested in humans. For example, dietary emulsifiers have been shown to reduce intestinal mucus layer thickness, increase microbial encroachment into the mucus layer, and increase susceptibility to metabolic syndrome and colitis [46]. This discovery was subsequently tested in a human trial of 16 subjects receiving either the common emulsifier carboxymethylcellulose cellulose CMC (*n* = 7) or an emulsifier-free diet (*n* = 9). Subjects receiving CMC had modestly increased postprandial abdominal discomfort with increased microbial encroachment into the mucous layer in 2 of 7 subjects in the CMC group [47]. The challenge with human studies is that the interpersonal variation in microbiome composition has a more significant effect than the differences observed in response to a dietary perturbation [48]. 

## 3. Prebiotics

Clinical trials evaluating the impact of prebiotics on IBD are limited (Table 1); however, intervention trials using germinated barley foodstuff (GBF) [49,50], psyllium [51,52], inulin/FOS [53,54] and xylooligosaccharides (XOS) [55,56,57] show promise due to their ability to modulate immune responses and improve disease activity. Limitations of these trials include the heterogeneity in primary endpoints and clinical disease activity scoring systems used, lack of objective biomarkers such as histology, and small sample sizes.

Xylans are plant cell-wall polysaccharides that support microbial fermentation [55]. Cereal grains are rich in xylan. Humans do not contain enzymes that degrade xylans; therefore, dietary xylans pass through the small intestine to the large intestine. Gut microbes, such as those that belong to the phylum Bacteroidota and Bifidobacterium genus possess a highly specialized core set of polysaccharide-binding proteins, outer membrane transporters, and glycolytic enzymes in their genome with the ability to cleave large polysaccharides, such as xylan, into oligosaccharides, referred to as xylooligosaccharides (XOS) [56]. XOS has been proposed as an emerging prebiotic due to their ability to counter gut inflammation by increasing the recovery of beneficial *Lactobacillus*, *Bifidobacterium*, and *Firmicutes* cell populations in the gut microbiome and increasing the production of SCFAs [55]. To date, human studies supplementing XOS are lacking; however, an in silico study (computer modeling) examining carbohydrate metabolic capabilities from metagenome-assembled genomes (MAGs) obtained from healthy, and CD patients shows promise [57]. In both groups, MAGs of *A. muciniphila*, *Barnesiella viscericola* DSM 18177, and *Paraprevotella xylaniphila* YIT 11841 contained enzymes (glycosidases) specific to the degradation of xylans, promoting species with key metabolic functions, capable of cross-feeding other beneficial species often reduced in CD. Therefore, xylan supplementation may ameliorate dysbiosis in CD through these mechanisms. Information on the impact of supplementation with hemicellulose-derived oligosaccharides, such as xylan on the gut microbiome, is emerging [56]. Well-designed clinical trials in IBD and XOS supplementation are necessary for elucidating the potential benefits to the gut microbiome.

## 4. Novel Fiber Sources and “Candidate” Prebiotics

### 4.1. Resistant Starch

Resistant starch (RS) is a broad category of structurally complex starches resistant to digestive enzymes in the gastrointestinal tract [62]. RS reaches the colon, where the gut microbiota metabolizes RS into a wide range of metabolites. It has both insoluble and soluble fiber characteristics, as it is water-soluble and readily fermented; however, it lacks viscosity. Five types of RS (RS 1–5) have been described, and their definition is based on the starch surface microstructure and how it interacts with digestive enzymes [63] (Table 2). At this time, RS2, RS3, and RS4 are candidate prebiotics as they do not meet the full definition of a prebiotic due to the variability in outcomes between studies [63].

Dietary supplementation with RS has been shown to significantly affect the gut microbiome in healthy individuals [67]. RS has consistently been found to promote the enrichment of *Ruminococcus bromii* and *Bifidobacterium adolescentis* [68]. *R. bromii* is considered a keystone species in RS metabolism as it plays a beneficial role in the degradation of various sugars of various lengths, releasing acetate to cross-feed other species [68]. *B. adolescentis* plays a similar role to *R. bromii* as a primary degrader of RS sugars, releasing lactate instead [69]. *Eubacterium rectale*, *Bacteroides thetaiotaomicron,* members of the genera *Roseburia*, *Butyrivibrio,* and *Bifidobacterium* are considered secondary degraders of RS, as they capture the degradation and fermentation products of primary RS degraders such as *R. bromii and B. adolescentis* to produce butyrate [68,69].

Human interventional studies using RS in IBD patients are limited. A meta-analysis (*n* = 7 studies), with the majority focusing on UC patients (*n* = 6), demonstrated that RS maintained clinical remission (based on disease activity scores), reduced the severity of symptoms in patients with active disease, and increased SCFA production, particularly butyrate [67]. RS 1 was most commonly investigated (*n* = 4), RS 2 (*n* = 1), a blend of RS1 and RS2 (*n* = 1), and a blend of RS2 and RS3 (*n* = 1) from high-amylose maize starch, oat bran, potatoes, and bananas. The intervention doses ranged from 0.6 g/day to 34.8 g/day, with a study duration of 5 days to 24 weeks. A limitation of existing clinical studies examining RS is the lack of objective markers of disease activity (e.g., histological and mucosal markers) and disease activity scoring tools, as each study used a different disease activity index. The estimated intake of RS by patients with IBD is 2.9 g/day (IQR 2.1–4.8 g/day), significantly less than the proposed recommendations of 20 g/day for gut health [70,71]. Although RS is a naturally occurring product and is likely to be safe [72], further research into the safety/tolerability and clinical efficacy of RS in an IBD population needs to be determined.

### 4.2. Polyphenols

Primary metabolites include lipids, carbohydrates, proteins, nucleic acids, and all essential elements required for cell growth and development [73]. Secondary metabolites are biologically active small molecules in specific cells that are not required for viability and provide a competitive advantage to the producing organism [73]. Polyphenols are secondary plant metabolites in plant-based foods such as fruits and vegetables, coffee and tea, whole grains, nuts, and legumes (Figure 3) [74]. There is high variability in polyphenol intake in the general population as intake depends on dietary habits [74]. For example, plant-based dietary patterns, such as the Mediterranean diet, which is rich in phenolic compounds, are estimated to provide 1 g/day versus 100–150 mg/day found in the Western dietary pattern, which is low in fruit and vegetables [75].

The protective role of polyphenols in health and disease is well-recognized due to their abundance of antioxidants and phytochemicals [76] and their emerging benefits to the gut microbiome [77]. As polyphenols are structurally diverse compounds, and their structure dictates their biological activity, the health effects will vary greatly [76,77]. Although the mechanisms of action of polyphenols are yet to be fully elucidated in humans (Figure 3), the current hypothesis is that phenolic compounds modulate the microbiota, specifically stimulating the growth of health-associated bacteria inhibiting pathobionts exhibiting a prebiotic effect (e.g., *Eubacterium rectale*, *Lactobacillus* spp., *Bifidobacterium* spp., *F. prausnitzii*) [75]. Polyphenol-derived metabolites produced by the microbiota can influence the composition of the microbiota and alter signaling pathways, for example, down-regulate inflammatory pathways (i.e., NF-κB) and influence epithelial barrier function by influencing intestinal permeability (i.e., tight junction assembly) [75]. Polyphenols are a promising therapy in IBD, as they affect the important pathological mechanisms involved in the pathogenesis of the disease.

#### 4.2.1. Resveratrol

Resveratrol is a stilbenoid polyphenol found in grapes (red wine), berries, soybeans, peanuts, and pomegranates with a wide range of biological properties, particularly potent anti-inflammatory and antioxidant effects [78]. One of the challenges of resveratrol is its low water solubility, lack of chemical stability, low bioavailability, and rapid metabolism; therefore, supplementation is needed to reap the beneficial health effects [78]. There is limited evidence to support the therapeutic effects of resveratrol treatment in humans with IBD [79,80,81,82]. An RCT that investigated the effects of 500 mg resveratrol or a placebo capsule for six weeks in active mild to moderate UC (*n* = 49) demonstrated that resveratrol supplementation could decrease the clinical activity index score, hs-CRP, tumour necrosis factor alpha (TNF-α) and inhibit the activity of NF-kB pathway [79]. A follow-up mechanistic study showed that resveratrol could reduce the disease activity score (SCCAI) and improve quality of life (IBDQ-9) in patients with UC through reduced oxidative stress and increased body antioxidant capacity [80]. Randomized controlled studies are required to validate the efficacy of resveratrol against inflammation and IBD treatment and to enhance our understanding of the mechanisms of action. 

#### 4.2.2. Curcumin

*Curcuma longa*, commonly known as turmeric, is a plant belonging to the Zingiberaceae family and is native to India and Southeast Asia. Turmeric contains compounds called curcuminoid pigments and polyphenols with important medicinal properties [83]. The actions of curcumin are achieved through inhibition of the NF-kB pathway, reducing the expression of interleukin (IL)-1, IL-6, IL-12, and TNF-α [84]. Commensal microbes, such as *Escherichia coli* CurA and *Vibrio vulnificus* CurA, also have enzymes that can convert curcumin into tetrahydrocurcumin, the major metabolite of curcumin responsible for its anti-inflammatory effects [85,86].

Evidence indicates that curcumin is an effective therapy for maintaining remission in UC when administered as a complementary therapy to mesalamine. Several meta-analyses have found that curcumin supplementation with mesalamine significantly improves clinical and endoscopic remission in UC [83,84,87,88]. Treatment doses ranging from 2–3 g/day have demonstrated the best efficacy. Curcumin has also been shown to be well tolerated and not associated with severe side effects. 

The evidence for curcumin supplementation In CD is not as robust, with only two RCTs published to date [89,90]. Patients with CD (*n* = 62) who underwent bowel resection treated with azathioprine (2.5 mg/kg) and were randomly assigned to receive oral curcumin (3 g/day; *n* = 31) or an identical placebo (*n* = 31) for six months. In this study, curcumin was no more effective than a placebo in preventing CD recurrence (Crohn’s disease activity index (CDAI) >150, Rutgeerts index ≥ i2a) [89]. Of note, this was a high-risk cohort, with 62.9% of the study arm having a postoperative reoccurrence. A highly absorbed curcumin (Theracurmin^®^, Europhartec, Lempdes, France) equivalent to 360 mg curcumin daily was administered to patients with active mild-to-moderate Crohn’s disease (CDAI < 150) for 12 weeks [90]. Theracurmin^®^ demonstrated significant reductions in clinical and endoscopic efficacy (Simple Endoscopic Score for Crohn’s Disease ≤ 4) from baseline to week 12 in the treatment group.

#### 4.2.3. Quercetin

Quercetin is a flavanol found in various foods, including apples, berries, capers, onions, and shallots [91]. Ingested in the form of glycosides (quercetin glycosides), the glycosyl groups are released during chewing, digestion, and absorption. Afterward, quercetin glycosides are hydrolyzed into aglycone via oral and gut microbes through the action of β-glycosidases enzymes before absorption into the enterocytes [91]. The therapeutic effects of quercetin in IBD are demonstrated to be through strengthening the integrity of the intestinal mucosal barrier, immune regulatory function, enhancing the diversity of colonic microbiota, and repressing oxidative stress [92].

A meta-analysis examining the effects of quercetin supplementation in preclinical models of IBD (11 animal studies with 199 animals) established that quercetin could improve histological scores, disease activity scores, inflammatory biomarkers (e.g., IL-1β, IL-10, TNF-α, and myeloperoxidase activity), and markers of oxidative stress (e.g., malondialdehyde, glutathione, superoxide dismutase activity, and catalase activity) [93]. We are unaware of any clinical trials that have examined the effect of quercetin in IBD; however, a clinical trial in healthy individuals with dysbiosis showed a marked reduction in various inflammatory markers such as IL-6, IL-1β, TNF-α, and lowered oxidative markers with quercetin supplementation [94]. Promising results from this clinical trial and preclinical trials suggest a possible role for quercetin in IBD.

### 4.3. Conjugated Linoleic Acid

Conjugated linoleic acid (CLA) is a class of positional and geometric isomers of polyunsaturated fatty acid linoleic acids [95]. Naturally occurring in the diet, CLAs are predominantly found in meats and milk from ruminant species produced through a chemical process called biohydrogenation [95]. To date, dietary requirements have not been defined, but intakes of North Americans are estimated to range from 152 to 212 mg for omnivores [96]. Approximately 90% of dietary intake of CLA intake is derived from meat and dairy products; however, the gastrointestinal microbiota, specifically *Bifidobacterium* spp., are also able to convert linoleic acid to CLA, making CLA a “candidate prebiotic” [97]. 

CLA has been studied for its potential health-promoting properties, including its effects on weight loss, food and energy intake, body composition, cancer, enhancement of immune function, and inflammation [98]. In preclinical models of colitis, CLA has been shown to attenuate colitis through the activation of PPAR-γ, an important negative regulator of inflammatory responses [95]. The immune functions of CLA have been examined in a small open-label trial (*n* = 13) of patients with mild to moderately active CD [99]. A dose of CLA (6 g/day) for 12 weeks significantly suppressed the ability of peripheral blood CD4+ and CD8+ T cell subsets to produce interferon (IFN)-γ, TNF-α, and IL-17, as well as significantly reduced clinical disease activity assessed by the CD activity index. The therapeutic benefit of CLA in IBD must be confirmed by more extensive placebo-controlled studies that examine the interaction between CLA supplementation, gut microbiota, and mucosal immunity.

## 5. Current Limitations in Our Understanding of Fiber in IBD

Advances in bacterial culturing and metabolomics have the potential to identify the metabolic outputs of specific bacterial strains in high throughput [100,101]. Despite the large number of rodent studies examining the impact of dietary fiber on the gut microbiome and metabolome, human studies are limited. Diet can influence many metabolites; therefore, it is unlikely that beneficial health effects are linked to a single metabolite. A consortium of metabolites is more likely to interact with the microbiota to produce beneficial health effects. Finally, although fiber is generally thought to be beneficial for the gut microbiome in healthy individuals, there is very little understanding of how fiber affects the microbiome of individuals with IBD [102]. There is limited evidence from intervention studies in IBD that IBD-associated dysbiosis can be modified in inactive and mild-moderately active diseases; however, tolerability varies greatly depending on current disease activity and fiber source. Interestingly, there is a suggestion that fermentation patterns are altered in IBD compared to healthy controls due to the altered functional capacity of the microbiome [103]. Of particular concern, a recent ex vivo study using colonic biopsies of patients with IBD found that unfermented inulin and FOS induced pro-inflammatory cytokines in a select group of patients, ultimately promoting inflammation [104]. This highlights that fiber supplementation may not be as benign as once thought. Individual microbiomes play a strong role in determining the outcomes and necessitate a more personalized nutritional approach to implementing dietary changes, including fiber supplementation (Figure 4). 

## 6. Opportunities for Personalized Nutrition 

Personalized nutrition, which aims to develop nutrition recommendations based on an individual’s intrinsic and extrinsic factors, holds great promise in managing diseases such as IBD. The responsiveness of the gut microbiota (including responders and effectors of host responses) appears to be largely dependent on baseline microbiota diversity and the specific microbes present or absent at baseline [105]. Machine learning approaches, which integrate and learn various patterns from datasets and discover predictive algorithms, are increasingly used to predict diet responses through the gut microbiome [106]. This approach has not yet been applied in IBD; however, it could be a valuable tool to predict an individual’s response to dietary fiber. For example, machine learning could be used to improve our ability to predict which microbes are responsive to various MACs and which ecological guilds work together to break down complex carbohydrates [107]. Novel tools such as glycan utilization screens can identify groups of bacteria capable of digesting fiber components [108]. The use of synthetic fibers might provide the opportunity to more precisely shape the strains targeted and metabolites generated by non-digestible dietary components [109]. Finally, researchers rely on food records which are very limited as current nutritional databases do not have the capability to capture the broad classes of fiber accurately. As the quality and quantity of input data are important for machine learning approaches, well-validated dietary collection and analysis methods to capture fiber intake are required. Exploring new and improved candidate biomarkers that reflect dietary fiber intake and dietary patterns is a potential solution.

## 7. Summary and Concluding Remarks

As the fiber pendulum swings, its use as a dietary therapy to improve outcomes in IBD holds promise; however, adequately powered randomized trials with objective clinical outcomes are urgently needed. As the inter-subject variability of the gut microbiota’s response to diet is likely the result of complex community interactions, an enhanced understanding of these interactions can inform the future design of precise diets that could potentially lead to improved outcomes for the IBD population. 

## Figures and Tables

**Figure 1 nutrients-15-01080-f001:**
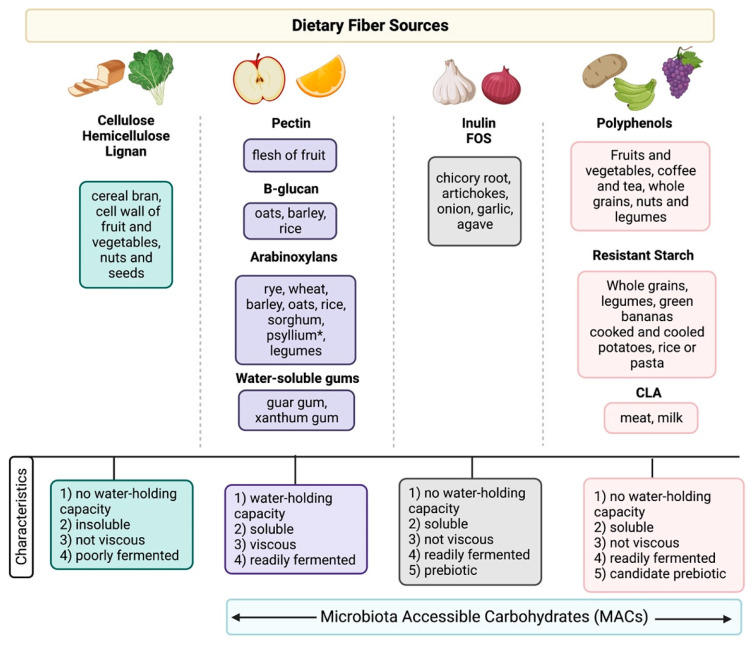
Sources of dietary fiber in the diet and classification based on physical characteristics (water-holding capacity, viscosity, and fermentation properties). Abbreviations: FOS, fructooligosaccharides; CLA, conjugated linoleic acid. * Psyllium is not well-fermented but has a water-holding capacity and is viscous. Figure made with Biorender.com (accessed on 20 January 2023).

**Figure 2 nutrients-15-01080-f002:**
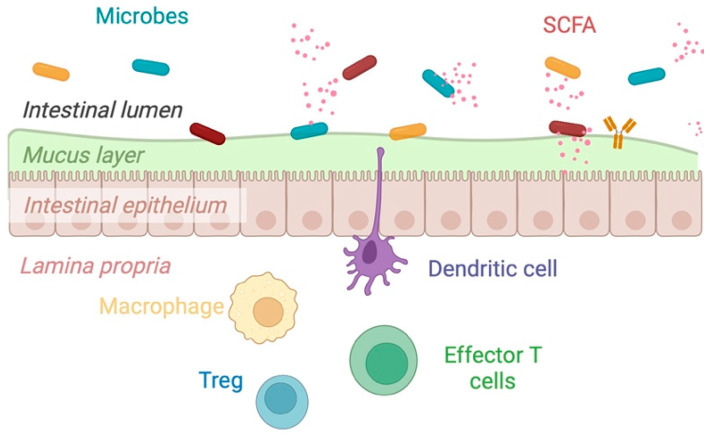
The tripartite relationship between the microbiota, the epithelial barrier, and the immune system. Homeostasis is maintained when there is eubiosis, a thick mucosal layer that protects the epithelial barrier, and balanced inflammation. (Image used with permission, N. Haskey “The Mediterranean diet pattern as a therapeutic approach for colitis,” Thesis, UBC, 2022); Created with Biorender.com (accessed on 31 January 2023).

**Figure 3 nutrients-15-01080-f003:**
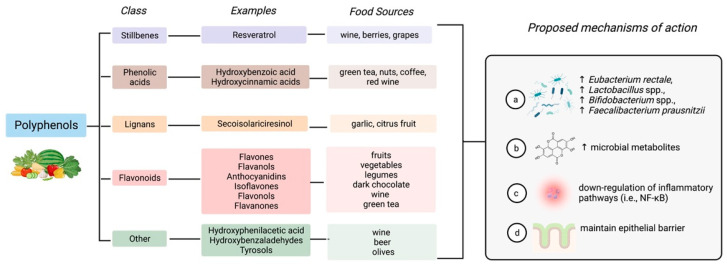
Polyphenols are a broad class of structurally diverse compounds found in plant-based foods that can affect the gut microbiome. Created with Biorender.com (accessed on 30 January 2023).

**Figure 4 nutrients-15-01080-f004:**
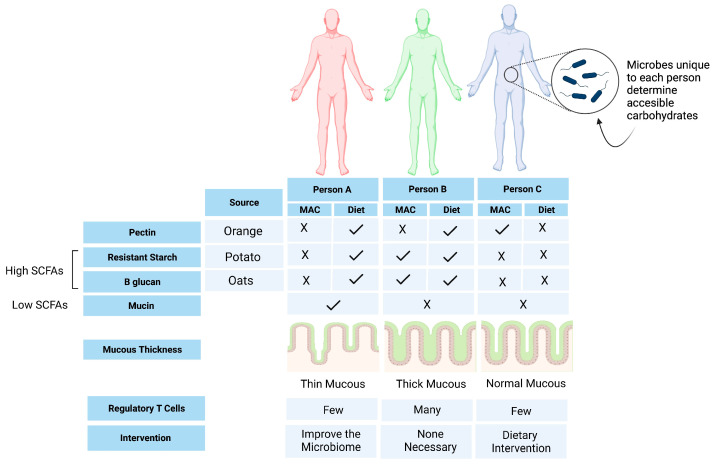
The unique set of bacterial strains harboured in each person’s gut microbiota reflects the microbiota-accessible carbohydrates (MACs) in their habitual diet. Likewise, the dietary fiber ingested by an individual determines which host-inaccessible carbohydrates are available to their gut microbiota to produce metabolites that influence health. Future interventions to prevent or treat IBD would be a combination of replacing missing microbes to increase beneficial metabolites from MACs, adjusting fiber consumption to provide the relevant substrates to the gut microbiota, or both. In this example, person A consumes a fiber-rich diet but lacks the microbes necessary to benefit the host; they instead harbour microbes that forage on mucus leading to a thinner mucus layer and increased susceptibility to inflammation through microbial access to the intestinal epithelium. For person A, a microbiome intervention to displace mucin-foraging bacteria and add microbes to maximize the benefits of their fiber-rich diet would be helpful. Person B has a good balance of fiber consumption and microbiota that complements their consumed fiber sources (resistant starch and beta-glucan), while person C has a MAC (pectin) but does not consume the pectin necessary to derive a health benefit and would therefore benefit from a dietary intervention. Created with Biorender.com (accessed on 20 January 2023).

**Table 1 nutrients-15-01080-t001:** Prebiotic intervention trials completed in IBD.

Author, Year	Study Design (*n* = 10)	Disease Type and Status	Intervention	Primary Endpoint	Results
Kanauchi et al. (2002) [49]	Open-label trial (*n* = 18)	Active UC (mild-moderate)	Standard medical therapy (control) or Standard medical therapy + GBF (20–30 g/day) for 4 weeks	Response to treatment measured by clinical disease score (Lichtiger method)	After 4 weeks of GBF administration, the clinical disease score in the GBF group was significantly lower than in the control group (*p* < 0.05).
Faghfoori et al. (2011) [50]	Randomized control trial (*n* = 41)	UC in remission	GBF (30 g/day) + standard medical therapy or standard medical therapy (control)	Changes to pre-treatment and post-treatment values of serum TNF-a, IL-6 and IL-8	Serum IL-6 and IL-8 decreased significantly in the GBF-treated group (*p* = 0.034 and *p* = 0.013). A trend towards increased TNF-α was seen in the non-GBF treated group (*p* = 0.08)
Casellas et al. (2007) [53]	Randomized, placebo-controlled trial (*n* = 15)	Active UC (mild-moderate)	Oligofructose-enriched inulin (12 g/day) + mesalazine (3 g/day) or placebo + mesalazine (3 g/day) for 2 weeks (control)	Reduction in inflammation as measured by fecal calprotectin and human DNA in feces	Oligofructose-enriched inulin plus mesalazine was associated with reduced fecal calprotectin (day 0: 4377 ± 659 ug/g; day 14: 1211 ± 449 ug/g, *p* < 0.05) but not in the placebo group. No changes were observed to DNA in feces in either group
Wilson et al. (2021) [58]	Open-label trial (*n* = 17)	Active UC (mild)	GOS (2.8 g/day) for 6 weeks	Changes in expression of any immune-related gene using a microarray of all genes expressed in the peripheral blood	No significant differences In immune gene expression were detected No change in disease activity, however a significant reduction in loose stools (*p* = 0.048) and urgency (*p* = 0.011) was observed No change in *Bifidobacterium*
Benjamin et al. (2011) [59]	Double-blind, placebo-controlled trial (*n* = 103)	Active CD	Oligofructose/inulin (15 g/day) or placebo for 4 weeks	Clinical response at week 4 (decrease in CDAI of ≥ 70 points)	No significant difference in the number of patients achieving a clinical response between the oligofructose/inulin and placebo groups (12 (22%) vs. 19 (39%), *p* = 0.067) Oligofructose/inulin had reduced proportions of interleukin (IL)-6-positive lamina propria DC and increased DC staining of IL-10 (*p* < 0.05) No change in fecal concentration of Bifidobacterium and *F. prausnitzii*
Hedin et al. (2021) [60]	Open-label trial (*n* = 19)	CD in remission	Oligofructose/inulin (15 g/day) for 3 weeks	Reduction in fecal calprotectin	Fecal calprotectin did not significantly change (*p* = 0.08) Fecal concentrations of Bifidobacteria and *Bifidobacterium longum* increased There was a significant reduction in intestinal permeability between baseline and following oligofructose/inulin supplementation in patients (urinary lactulose-rhamnose ratio from median 0.066, IQR 0.092 to median 0.041, IQR 0.038, *p* = 0.049)
Joossens et al. (2011) [61]	Randomized, placebo-controlled trial (*n* = 67)	Inactive to mild to moderately active CD	10 g/day oligofructose/inulin twice daily for one month	Reduction in disease activity measured by Harvey-Bradshaw Index (HBI) and changes to the microbiota	A significant increase in *B. longum* was seen in the treatment (ITT *p* = 0.03) Sub-group analyses revealed a significant decrease in HBI from 7 to 5 following treatment (*p* = 0.03)
Lindsay et al. (2006) [54]	Open-label trial (*n* = 10)	Active ileocolonic CD	15 g/day oligofructose/inulin for 3 weeks	Reduction in disease activity measured by Harvey-Bradshaw Index (HBI)	There was a significant reduction in HBI from baseline 9.8 (SD 3.1) to 6.9 (3.4) at week 3 (*p* = 0.01) FOS induced a marked increase in fecal *Bifidobacteria* concentrations (baseline 8.8 (0.9) to FOS 9.4 (0.9) log cell/g dry feces; *p* = 0.005)
Fernández-Bañares et al. (1999) [51]	Open-label, parallel-group, randomized controlled trial (*n* = 94)	UC in remission	*Plantago ovata* seeds (20 g/day), mesalamine (1500 mg/day), or *Plantago ovata* seeds (20 g/day) + mesalamine (1500 mg/day) for 12 months	Maintenance of remission at 12 months	The treatment failure rate was 40% (14/35 patients) in the *Plantago ovata* seed group, 35% (13/37) in the mesalamine group, and 30% (9/30) in the *Plantago ovata* seed plus mesalamine group. The probability of remission was similar between groups (*p* = 0.67)
Hallert et al. (1991) [52]	Double-blind, placebo-controlled, crossover trial (*n* = 29)	UC in remission	Ispaghula husk (4 g twice daily) or placebo for 2 months each	Reduction in gastrointestinal symptoms: abdominal pain, diarrhea, loose stools, urgency, bloating, mucus, incomplete evacuation, constipation	Ispaghula husk was consistently superior and associated with a significantly higher rate of improvement (69%) in gastrointestinal symptoms than placebo (24%) *(p* < 0.001)

Abbreviations: UC: ulcerative colitis; CD: Crohn’s disease; GBF: germinated barley foodstuff; IL: Interleukin; ITT: intention to treat; TNF-α: tumour necrosis factor alpha; GOS: galactooligosaccharides; IQR; interquartile range. Novel Fiber Sources and “Candidate” Prebiotics.

**Table 2 nutrients-15-01080-t002:** Types of Resistant Starch.

RS Types	Definition	Food Sources	Reference
RS 1	Physically inaccessible starch found entrapped within the protein matrix or non-starch components of the plant cell wall	Unprocessed whole grains, legumes such as soybean seeds, beans, lentils, and dried peas	Li et al., 2021 [64]
RS 2	Resistant starch granules	Raw potato, green banana, high amylose cornstarch	Li et al., 2021 [64]
RS 3	Obtained by retrogradation process upon cooking and cooling of starch-containing foods	Cooked or cooled rice, pasta or potatoes, cornflakes	Topping et al., 2003 [65]
RS 4	Starch-modified through chemical processes	Food additives derived from corn, potatoes, or rice are used for formulations that require smoothness, pulpy texture, flowability, low-pH storage, and high-temperature storage	Fuentes-Zaragoza et al., 2011 [62]
RS 5	Starch obtained by complex formation between high amylose starch with the lipids	Resistant maltodextrin, high-amylose starch	Hasjim et al., 2013 [66]

## Data Availability

Not applicable.

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
