# Peer review of "To Fiber or Not to Fiber: The Swinging Pendulum of Fiber Supplementation in Patients with Inflammatory Bowel Disease"

_nutrients, 2023, doi:10.3390/nu15051080_

Round 1
Reviewer 1 Report
1. Introduction:It seems abrupt to use only one reference [5] as the entry point of the review paper. Moreover, it is not enough to only consider the type, form and quality of DF. The specific situation of IBM patients should also be considered.
2. Line 118. “the optimal fiber source, both type and quantity, is yet to be defined” The optimal fiber source may be hard to find. Can the manuscript rephrase it?
3. Line 119-135. The intention of this section is not clear, improve it please!
4. Line 167-180. Just an overview of what others have reported, it would be nice if the manuscript could summarize the positive and negative strains from A great deal reports for IBD!
5. Line 194-195. Judging from the text of this passage, the bacteria seem to be bad ones, and they are supposed to be good bacteria! Following up references [33], it was found that the experimental method adopted in this report was to enrich mucophilic bacteria in adult intestinal flora by using mucin as carbon and nitrogen sources. Remember that Akkermansia muciniphila can stimulate mucin production in goblet cells as well as break down mucin! Suggest changing the way of expression!
6. Line 201-213. Please don't just give the expression of some reports and be done with it! The recognition relationship between dietary fiber and cell specific receptors and the effect of induction is more important!
7. Line 235-241. This part of the content is too little, suggest supplement! Some of the ingredients (such as XOS) that follow should also be prebiotics. In a way, the headings would better be re-arrangement!
8. About 4.1 Resistant starch: After all, resistant starch is not a natural product. It is recommended not to put it in the first place and occupy too much space.
9. About 4.2 Polyphenols: This part of the content is more scattered, it is suggested to summarize again!
10. About 6. Conjugated Linoleic Acid: It's not supposed to be fiber! These contents may not be relevant to the title of the manuscript, it is recommended to delete!
11. About 7. Fiber Conundrums: Please get out the subjects (intention) directly, not too indirect!
12. About 9. Summary and Concluding Remarks: This expression seems to leave something to be desired. Here, if the manuscript could give clear and conclusive suggestions in the aspects of "mechanism" and "precision nutrition", farther, to echo the "prediction" in the abstract part, it will be meaningful!
Reviewer 2 Report
This is an extensive narrative review on the clinical effects of fiber in IBD patients. I have read the article with great pleasure and interest. It wouldn't look out of place as a chapter in a medical textbook. I think it may be suitable for publication in Nutrients. However, there are some points that need attention. Adjusting these issues may improve the quality of the review even further
·The quality, composition and food technology contribute to the clinical effect of the dietary fiber. In this context, it is interesting to explain and discuss the role of intrinsic fiber a little more in this extensive review. (Puhlmann ML, de Vos WM. Intrinsic dietary fibers and the gut microbiome: Rediscovering the benefits of the plant cell matrix for human health. Front Immunol. 2022 Aug 18;13:954845. doi: 10.3389/fimmu.2022.954845. PMID: 36059540; PMCID: PMC9434118).
·The title is about fiber and fiber alone. In the review however, also other nutrients and supplements are mentioned. Although the stimulating title is very appealing, it does not cover the entire load of the review.
·Line 41-42: …produce important microbolites. /It’s better to name them/.
·Line 54: havingvarying ïƒ /having varying./
·Line 108: ….owing to insufficient reproducibility in intervention trials. /Can you add one or two references? /
·Line 229-230: receiving either the common emulsifier carboxymethylcellulose /cellulose/ CMC (n=7) or an emulsifier-free diet (n=9)
·Table 1: /is it possible to give a quality assessment of the selected studies? This will give the reader more insight on how to interpret the results of the studies./
·Line 277-279: Although RS is a naturally occurring product and is likely to be safe [61], further research into the safety/tolerability of RS in an IBD population needs to be determined. /I think that the clinical effectivity also needs further determination?/
//
·Line 341: …clinical and endoscopic remission in UC [72]. [73][76][77].
